# N-Type Ca Channel in Epileptic Syndromes and Epilepsy: A Systematic Review of Its Genetic Variants

**DOI:** 10.3390/ijms24076100

**Published:** 2023-03-23

**Authors:** Sonia Mayo, Irene Gómez-Manjón, Ana Victoria Marco-Hernández, Francisco Javier Fernández-Martínez, Ana Camacho, Francisco Martínez

**Affiliations:** 1Genetics and Inheritance Research Group, Instituto de Investigación Sanitaria Hospital 12 de Octubre, 28041 Madrid, Spain; 2Department of Genetics, Hospital Universitario 12 de Octubre, 28041 Madrid, Spain; 3Neuropediatric Unit, Hospital Universitario Doctor Peset, 46017 Valencia, Spain; 4Translational Research in Genetics, Instituto de Investigación Sanitaria La Fe, 46026 Valencia, Spain; 5Division of Pediatric Neurology, Department of Neurology, Hospital Universitario 12 de Octubre, Universidad Complutense de Madrid, 28040 Madrid, Spain; 6Genomic Unit, Instituto de Investigación Sanitaria La Fe, 46026 Valencia, Spain; 7Genetics Unit, Hospital Universitario y Politecnico La Fe, 46026 Valencia, Spain

**Keywords:** N-type Ca channel, *CACNA1B*, epilepsy

## Abstract

N-type voltage-gated calcium channel controls the release of neurotransmitters from neurons. The association of other voltage-gated calcium channels with epilepsy is well-known. The association of N-type voltage-gated calcium channels and pain has also been established. However, the relationship between this type of calcium channel and epilepsy has not been specifically reviewed. Therefore, the present review systematically summarizes existing publications regarding the genetic associations between N-type voltage-dependent calcium channel and epilepsy.

## 1. Introduction

### 1.1. Voltage-Dependent Calcium Channels

Voltage-gated calcium channels (VGCCs) mediate Ca^2+^ influx in response to action potentials and subthreshold depolarizing signals. They participate in the signal transduction of many physiological processes, such as contraction, synaptic transmission, hormone secretion, enzyme activity, or gene expression [1]. At the molecular level, VGCCs are comprised of α1, α2/δ, β, and γ subunits (Figure 1). The α1 subunit is the ion-conducting element and contains the gating and voltage sensor machinery of the channel [2]. Ten different α1 subunits have been characterized and defined the types of these channels, which have been classified into two groups: high-voltage activated (HVA), when open at relatively depolarized membrane potentials, and low-voltage activated (LVA) Ca^2+^ channels when opening near the resting membrane potential (Table 1) [2,3,4]. The biophysical and pharmacological properties of these channels are included in Table 2. The other subunits modulate and regulate the α1 pore-forming subunit: The β-subunit from the cytoplasm, the α2δ subunit from the extracellular matrix, and the γ subunit, not always present, that seems to downregulate the channel activity [1,5,6] (Table 3).

So far, the association of some voltage-gated calcium channel components with epilepsy has been widely reported. For instance, some pathogenic variants in *CACNA1A* and *CACNA1E* genes cause developmental and epileptic encephalopathies [MIM# 617106 and MIM# 618285]. In other cases, this association remains controversial. For instance, although different variants in *CACNA1H* have been reported in patients with congenital forms of idiopathic generalized epilepsies (IGEs) or absence epilepsy [7,8,9], their classification as pathogenic variants and the association of *CACNA1H* with epilepsy has been questioned [10]. The relationship between N-type voltage-gated calcium channels and epilepsy has not been specifically reviewed.

**Figure 1 ijms-24-06100-f001:**
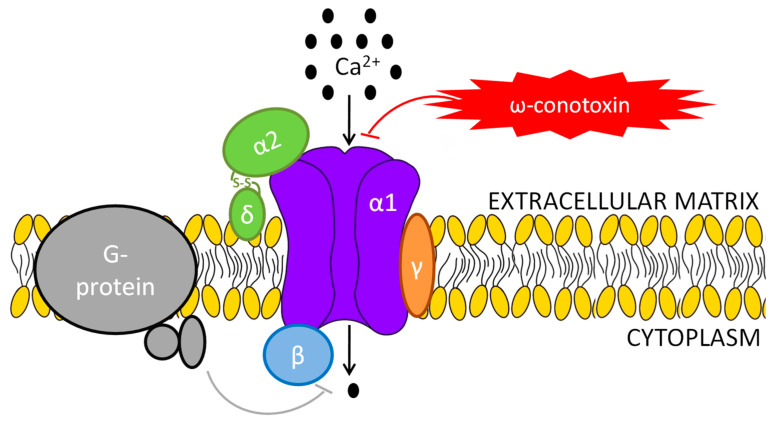
Schematic representation of the structure of the voltage-gated Ca^2+^ channels.

**Table 1 ijms-24-06100-t001:** Voltage-gated Ca^2+^ channel types according to the α1 subunit.

Channel Type	Channel Name	Gene (Pore-Forming Subunit)	*Locus*	LVA/HVA
P/Q	Cav2.1	*CACNA1A*	19p13.13	HVA
N	Cav2.2	*CACNA1B*	9q34.3	HVA
L	Cav1.2	*CACNA1C*	12p13.33	HVA
L	Cav1.3	*CACNA1D*	3p21.1	HVA
R	Cav2.3	*CACNA1E*	1q25.3	HVA
L	Cav1.4	*CACNA1F*	Xp11.23	HVA
T	Cav3.1	*CACNA1G*	17q21.33	LVA
T	Cav3.2	*CACNA1H*	16p13.3	LVA
T	Cav3.3	*CACNA1I*	22q13.1	LVA
L	Cav1.1	*CACNA1S*	1q32.1	HVA

HVA: high-voltage activated, LVA: low-voltage activated.

**Table 2 ijms-24-06100-t002:** Biophysical and pharmacological properties of voltage-gated Ca^2+^ channels.

Channel Type	Conductance (pS)	Activation Potential (mV)	Inactivation Kinetics (ms)	Specific Blocker
L	25	~10–50	150–2000 (slow)	dihydropyridines
N	11–20	~20	100–200 (intermediate)	ω-conotoxin
P/Q	15–16	~50	500–1000 (intermediate–slow)	ω-Agatoxin
R	15–20	~25–40	50–100 (intermediate–fast)	SNX-482
T	8	~70	10–70 (fast)	Mibefradil

Adapted from Gurkoff et al. (2013) [11].

**Table 3 ijms-24-06100-t003:** Genes coding for the different auxiliary subunits of the voltage-gated Ca^2+^ channels (β, α2δ, and γ).

Gene	Subunit Name	*Locus*	OMIM *
*CACNB1*	β1	17q12	114207
*CACNB2*	β2	10p12.33-p12.31	600003
*CACNB3*	β3	12q13.12	601958
*CACNB4*	β4	2q23.3	601949
*CACNA2D1*	α2δ1	7q21.11	114204
*CACNA2D2*	α2δ2	3p21.31	607082
*CACNA2D3*	α2δ3	3p21.1-p14.3	606399
*CACNA2D4*	α2δ4	12p13.33	608171
*CACNG1*	γ1	17q24.2	114209
*CACNG2*	γ2	22q12.3	602911
*CACNG3*	γ3	16p12.1	606403
*CACNG4*	γ4	17q24.2	606404
*CACNG5*	γ5	17q24.2	606405
*CACNG6*	γ6	19q13.42	606898
*CACNG7*	γ7	19q13.42	606899
*CACNG8*	γ8	19q13.42	606900

* Gene description in OMIM. OMIM: Online Mendelian Inheritance in Man.

### 1.2. N-Type Ca Channel

N-type calcium channels are characterized by their exclusively neuronal tissue distribution, prominent inhibition by G-proteins, and their participation in nociception [12]. They are sensitive to ω-conotoxins, small neurotoxic peptides isolated from the venom of sea snails of the genus *Conus* [13], some of which have been used in functional assays and for pharmacological purposes [14].

The *CACNA1B* gene [MIM*601012] encodes for the pore-forming subunit of this channel (α1B), which is expressed throughout the central nervous system and in various brain regions, controlling neurotransmitter release from neurons. α1B can form different complexes with the auxiliary α2δ and β subunits to form the N-type Ca channels.

The α2δ subunits are membrane-bound extracellular glycoproteins abundantly expressed in the central and peripheral nervous system. The α2δ pre-proteins are post-translationally cleaved into two polypeptides, α2 and δ [15], which remain attached by disulfide bridges and extra-cellularly linked to the plasma membrane through a glycosylphosphatidylinositol (GPI) anchor attached to delta [16]. Four different α2δ subunits have been described in humans, coded by *CACNA2D1* through *CACNA2D4* genes. Different studies have pointed out the association of α2δ1 (coded by *CACNA2D1*) and α2δ2 (coded by *CACNA2D2*) with α1B at the plasma membrane [17]. However, an increasing number of studies suggest that the α2δ subunits may have additional functions independent of the calcium channel function. Thus, a model proposed that the uncleaved α2δ subunits maintain immature calcium channels in an inhibited state, while the proteolytic processing of α2δ leads to an activation of the channels, acting as a checkpoint for trafficking of mature calcium channel complexes into neuronal processes [18]. In support of this model, the α2δ1 subunit was proved to be of great relevance for membrane trafficking of the α1B subunit in some types of neurons [19].

The β subunits regulate the channel by modulating its gating and its interaction with other proteins and molecules, such as G proteins [20]. According to the quantitative analysis of Cav2 channel proteins in the mammalian brain, from the four β subunits, β4 is the most abundant auxiliary subunit of all Cav2 channel subtypes, including N-type [21]. This subunit is encoded by the *CACNB4* gene [MIM*601949]. In addition, the β3 is the second most frequent subunit assembled to α1B [21], and may be implicated in the association of the N-type channel with pain processing [22].

The auxiliary subunits α2δ1, α2δ2, and β4 may have different effects in the α1B subunit and coffered different properties to the N-Type channels [12].

## 2. Methods

Electronic research was conducted to identify pertinent English articles examining the genetic associations between the N-type voltage-dependent calcium channel and epilepsy. Two types of databases were consulted: PubMed for publications and ClinVar and HGMD professional for specific case reports (Figure 2). Therefore, studies with both clinical and preclinical information were included in this review.

A systematic literature research of PubMed was performed to identify eligible articles until 4 December 2022 using the following search terms: (epilepsy OR epileptic OR seizure) AND (N-type voltage-dependent calcium channel OR Cav2.2 OR CACNA1B). The search identified 166 potential articles.

Reviews, editorials, commentaries, and articles in a different language than English were excluded. We screened the titles and abstracts to check if they fell within the scope of this review. In some cases, when abstracts were not available or more information was required to decide, a quick review of the whole article was carried out. At this stage, a total of 34 original articles were selected.

For clinical cases also ClinVar and HGMD professional databases were revised. From 75 pathogenic variants for *CACNA1B* reported in ClinVar, 71 were copy number variants (CNVs) affecting more genes (35 deletions and 36 duplications), the other 4 variants were described by Gorman et al. [24] and included in the systematic review.

Regarding the auxiliary subunits, from 17 pathogenic variants for *CACNA2D1* reported in ClinVar, two were chromosomal rearrangements (a ring chr7 and an inversion); 12 were CNVs affecting more genes (6 deletions and 6 duplications), the other three variants were described by Dahimene et al. [25] and also included in the systematic review.

From 32 pathogenic variants for *CACNA2D2*, 9 were CNVs affecting more genes (3 deletions and 6 duplications), one was an intragenic deletion in a patient with epileptic encephalopathy and 22 were SNPs or frameshift variants (3 missense, 6 nonsense, 2 splicing variants, 11 frameshift) of which 20 were associated with epilepsy or seizures and 2 were not associated to a specific phenotype.

From 17 pathogenic variants for *CACNB4*, 16 were CNVs affecting more genes (7 deletions and 9 duplications), and one was a frameshift variant detected in a patient with idiopathic epilepsy.

In relation to HGMD professional, for *CACNA1B*, 9 deleterious variants were reported, but only 4, included in Gorman et al. [24], are associated with epilepsy. For *CACNA2D1*, 11 deleterious variants are described, of which 7, were not found in the systematic review and were associated with epilepsy: 4 CNVs (deletions) which affect more genes, a splicing variant [26], frameshift variant [27] and a balanced translocation that disrupted *CACNA2D1* [28]. For *CACNA2D2*, we also found 11 deleterious variants and 10 associated with epilepsy [29,30,31,32,33,34]. For *CACNB4*, 9 deleterious variants were found, 3 deletions affecting other genes, and the rest of the variants (SNP and frameshift) were associated with epilepsy [35,36,37,38,39]. For *CACNG2*, 4 deleterious variants were found, and only one was associated with epilepsy [40].

For *CACNB1*, *CACNB2*, *CACNB3*, *CACNA2D3*, *CACNA2D4*, *CACNG1*, *CACNG3*, *CACNG4*, *CACNG5*, *CACNG6*, *CACNG7*, and *CACNG8*, no pathogenic variants associated with epilepsy or seizures and affecting exclusively one of these genes were found neither in ClinVar nor HGMD.

## 3. Results and Discussion

### 3.1. N-Type Ca Channel and Epilepsy: Clinical Evidence

More than 25 epileptic patients with pathogenic variants in *CACNA1B* or other genes encoding auxiliary subunits of VGCC have been reported in the last eight years, pointing to, and indeed confirming, the association of N-type voltage-gated calcium channel with epilepsy (Table 4).

#### 3.1.1. α1 Subunit: *CACNA1B*

Bi-allelic loss-of-function *CACNA1B* pathogenic variants have been associated to neurodevelopmental disorder with seizures and nonepileptic hyperkinetic movements [MIM# 618497]. Until now, only six children from three unrelated families have been reported [24]. These patients presented a complex and progressive neurological syndrome, with epileptic encephalopathy between 9 and 30 months, severe neurodevelopmental delay, and with regression, among other features. Three sibs from a consanguineous family were homozygous for the variant c.3665del, (p.Leu1222Argfs*29); two sibs were compound heterozygous for the variants c.3573_3574del, (p.Gly1192Cysfs*5), and c.4857+1G>C; and an adopted child homozygous for the variant c.1147 C>T (p.Arg383Ter) in *CACNA1B*. Currently, it is necessary to further investigate the involvement of pathogenic variants in the *CACNA1B* gene in neurodevelopment and epileptic disorders, as well as to fully understand the etiopathogenic mechanisms of the disease.

#### 3.1.2. α2δ Subunits: *CACNA2D1* and *CACNA2D2*

##### *CACNA2D1* 

So far, only two unrelated patients have been reported with early-onset developmental epileptic encephalopathy due to biallelic variants causing a loss-of-function of the *CACNA2D1* gene [25]. Two frameshift variants were found, either homozygous or in compound heterozygosity with the missense variant p.(Gly209Asp), which abolishes the proteolytic maturation of α2δ and severely affects Ca_V_2 calcium channel function. Generalized seizures developed at age 11–19 months. Both patients showed severe hypotonia, as an initial manifestation before three months of age, microcephaly, absent speech, spasticity, choreiform movements, orofacial dyskinesia, and cortical visual impairment. Brain imaging revealed corpus callosum hypoplasia and progressive volume loss in both.

In addition, four patients with intellectual disability (ID) and generalized epileptic seizures were previously reported as carriers of different variants, which allowed the authors to propose this gene as a candidate gene for epilepsy and ID. A de novo balanced translocation that disrupted the *CACNA2D1* gene, a de novo 7.5 Mb deletion harboring 14 genes, and a 2.72Mb deletion detected in a patient and in her mother, who presented mild ID but not epilepsy, were initially published [28]. It is worth noting that the deletions contain other genes (*PCLO*, *SEMA3E*, *SEMA3A*), which could conceivably contribute to the phenotype, as they are involved in neuronal synapsis, axonal guidance, or development of the nervous system. Two of these patients additionally showed different inconsistent dysmorphic features; two of them associated short stature, and one patient revealed MRI findings (bilateral frontotemporal polymicrogyria, mega cisterna magna, and a cyst in the cavum veli interpositi). Intriguingly, the larger deletion was associated with the mildest phenotype in a girl with an IQ of around 80 who initially had an apparently normal neuromotor development but underwent a slight regression after several seizures. On the other hand, a de novo splicing variant was posteriorly reported in a fourth patient with epilepsy and ID [26], and a de novo frameshift variant was detected in a female with normal intelligence and infantile spasms that responded to ACTH with good outcome [27]. Currently, it is not established whether these loss-of-function variants in a heterozygous state cause intellectual disability and epilepsy. Taking into consideration the recessive pattern seen in the above patients and the fact that this gene shows some tolerance to loss-of-function variants [41], as well as some deletions observed in healthy individuals [42,43], one should be cautious about the relevance of these genetic variants. Maybe some heterozygous loss-of-function variants in this gene indeed predispose to epilepsy and/or ID with incomplete penetrance. However, they may also be casual findings or, in some cases, be present in combination with other unnoticed variants in trans in the *CACNA2D1* gene, which altogether contribute to the clinical manifestations.

**Table 4 ijms-24-06100-t004:** Summary of the clinical presentation of the reported cases with epilepsy and genetic alterations in the N-type Ca channel.

Ref.	Gene	Nº Patients	Epilepsy	Cognitive Involvement	MRI	Prenatal/ Birth	Other
[24]	*CACNA1B*	6	DEE	Yes (severe/regression)	Unknown	Normal	Hyperkinetic movement disorder, postnatal microcephaly, hypotonia. Death in childhood (5/6)
[25]	*CACNA2D1*	2	DEE	Yes (severe)	Atrophy, thin corpus callosum	Normal	Facial dysmorphism, hyperkinetic movement disorder, insensibility to pain
[28]	*CACNA2D1*	3	Focal epilepsy (2/3 refractory epilepsy)	Yes (mild to moderate)	1 normal, 1 atrophy in the follow-up,1 frontotemporal polymicrogyria	Normal	2 facial dysmorphism, 1 hyperinsulinism and obesity
[26]	*CACNA2D1*	1	Epilepsy (no additional data)	Yes	Unknown	Unknown	
[27]	*CACNA2D1*	1	DEE, infantile spasms	No	Normal	Unknown	Facial dysmorphism
[30]	*CACNA2D2*	3	DEE	Yes (severe)	2/3 paucity of white matter and cerebellar atrophy	Normal	Hyperkinetic movement disorder, hypotonia
[31]	*CACNA2D2*	1	DEE	Yes	Cerebellar atrophy	Normal	Estrabismus, ocular apraxia
[29]	*CACNA2D2*	1	DEE	Yes	Cerebellar atrophy	Fetal distress	Ataxia, hypotonia, atypical eye movements
[44]	*CACNA2D2*	1	1 febrile seizure	No (only motor delay)	Cerebellar atrophy		Ataxia, neonatal hypotonia, myoclonus
[33]	*CACNA2D2*	3	DEE	Yes (severe)	Cerebellar and brain atrophy	Normal	1 Facial dysmorphism, 1 ataxia. Death in childhood (1/3)
[35]	*CACNB4*	10	3/10 generalized idiopathic epilepsy	No	Unknown	Unknown	5/10 episodic ataxia, 2/10 asymptomatic
[36]	*CACNB4*	1	Severe myoclonic epilepsy in infancy	Yes	Normal	Unknown	Ataxia. Death in childhood
[37]	*CACNB4*	1	NA	NA	NA	NA	NA
[38]	*CACNB4*	2	1 DEE/1 focal epilepsy	Yes (severe)	Cerebellar atrophy	Normal	Hyperkinetic movement disorder, hypotonia
[40]	*CACNG2*	1	Benign rolandic epilepsy	No	Unknown	Unknown	

DEE: developmental and epileptic encephalopathy; NA: not available; Ref: references.

##### *CACNA2D2* 

Initial evidence of the phenotype associated with *CACNA2D2* pathogenic variants came from the mouse mutant strain ducky, which carried null alleles for Cacna2d2. These animals are a model for absence epilepsy characterized by behavioral arrest synchronous with spike-wave discharges and severe cerebellar ataxia. This reflects the strong expression of α2δ2 in particular neurons, specifically cerebellar Purkinje cells [45]. In humans, biallelic *CACNA2D2* variants cause a phenotypic spectrum ranging from congenital ataxia with cerebellar vermian atrophy on brain imaging [44] to cerebellar atrophy and developmental and epileptic encephalopathies [30,31]. The homozygous variant p.L1040P in the *CACNA2D2* gene was initially related to a dysfunction of α2δ2, resulting in reduced current density and slow inactivation in both N-type (Cav2.2) and L-type (Cav1.2) calcium channels. The resulting prolonged calcium entry during depolarisation and changes in the surface density of calcium channels was proposed to underly the epileptic phenotype [30]. Later, abolished expression of the *CACNA2D2* gene, due to a homozygous frameshift loss-of-function variant, in a patient was significantly associated with severe delays of psychomotor development, epilepsy with absence seizures, dyskinetic movements, and cerebellar atrophy [31]. Some dysmorphic features and microcephaly were also present. The identification of novel pathogenic or likely pathogenic variants in the homozygous state leads further support to the association of the *CACNA2D2* gene with developmental and epileptic encephalopathy with cerebellar atrophy and ataxia [29,33]. On the other hand, Valence et al. describe an individual with congenital ataxia and cerebellar atrophy, where the milder neurologic phenotype was attributed to the partial splicing defect of a rare homozygous splice variant [44]. These authors demonstrated a diminished expression of wild-type mRNA transcript, which would explain the milder neurologic conditions without ID and with a single febrile seizure episode.

Considerable clinical overlap between individuals with homozygous loss-of-function variants in *CACNA2D1* and in the *CACNA2D2* gene has been highlighted [25]. However, atrophy of the brain affects the cerebrum in individuals with *CACNA2D1* variants, whereas cerebellar atrophy was consistently reported in subjects with *CACNA2D2* variants, a difference most probably due to a higher expression of the *CACNA2D2* gene in cerebellar cells [33]. Similarly, the predominant expression of *CACNA2D4* in the retina leads to retinal dysfunctions in affected patients, which indicates that any of the other α2δ subunits cannot compensate for the loss of α2δ1 or α2δ2 during development [25].

#### 3.1.3. β Subunit: *CACNB4*

Mutations in the *CACNB4* gene might confer susceptibility to idiopathic generalized epilepsy-9 (IGE9) and juvenile myoclonic epilepsy-6 (JME6) [MIM# 607682]. Taking into account that the lethargic (lh) mouse, who has inactivated the *Cchb4* gene, present a complex neurological disorder that includes absence epilepsy and ataxia [46]; Escayg et al. screened in 90 familial with IGE and 71 families with episodic ataxia for mutations in *CACNB4* genes [35]. Two coding changes that were considered probably pathogenic were identified in three different families. A female patient with juvenile myoclonic epilepsy (JME) was heterozygous for the premature-termination mutation c.1444C>T (p.Arg482Ter). In addition, functional analysis in *Xenopus laevis* oocytes co-expressing α1A and β4 subunits indicated that p.Arg482Ter might have altered the kinetic of inactivation [35]. Besides, this variant may alter other functions of CACNB4 as a repressor of neuronal gene expression [47]. The other variant, c.311G>T (p.Cys104Phe), was identified in nine individuals from two unrelated families. Two affected individuals with an atypical but similar clinical syndrome of IGE with rare juvenile atypical prolonged absences and occasional GTCS from one family; and from the other family, two asymptomatic individuals, four relatives with a more heterogeneous phenotype of ataxia and/or vertigo, and one case with febrile seizures as a child [35]. However, according to ClinVar (Variation ID:7608) and based on different submitters, the pathogenicity of this variant is unclear.

In a patient heterozygous for the pathogenic variant c.1702C>T (p.Arg568Ter) in *SCN1A*, de novo, with severe myoclonic epilepsy in infancy (SMEI), a second variant in *CACNB4*, was reported. The c.1403G>A (p.Arg468Gln) in *CACNB4*, inherited from his father, who had a febrile seizure associated with measles at the age of seven, was considered as a possible phenotype modifier [36].

In a cohort of 280 unrelated patients with early-onset developmental and epileptic encephalopathy (DEE) screened for alteration in 172 genes, the pathogenic variant c.21delC (p.Lys8Argfs*26) in *CACNB4* was identified, although specific clinical data from this case was not provided [37].

Finally, from a cohort of 390 subjects with neurodevelopment disorder, a homozygous variant in *CACNB4* was detected in two siblings from first-degree consanguineous healthy parents. Both patients present severe developmental delay and cerebellar atrophy with epilepsy, among other features. The younger brother presented focal seizures and responded to treatment, while the older sister, with a more severe phenotype, presented focal and tonic seizures refractory to treatment. In silico tools, animal models, and clinical and genetic data suggest the c.377T>C (p.Lys126Pro) *CACNB4* variant is likely pathogenic and impairs the formation of the calcium channel complexes. However, both siblings also carried 11 other rare homozygous variants, which could contribute to the severe phenotype [38].

Although some cases with epilepsy and alterations in *CACNB4* have been reported, the variability in the phenotype, the weak evidence of pathogenicity in some cases, and the presence of other genetic variants in others make it necessary to get further evidence to establish a stronger association between *CACNB4* with epilepsy.

#### 3.1.4. γ Subunit: *CACNG2*

The protein encoded by the *CACNG2* gene [MIM* 602911], expressed specifically in brain, is a transmembrane AMPA receptor regulatory protein (TARRP) and also the γ2 subunit of Ca^2+^ channels. γ2 inhibits Cav2.2 channels, downregulating its activity [48].

Only a patient with atypical Rolandic epilepsy has been reported with a de novo variant in *CACNG2* gene so far [40]. Without further evidence, the c.295+1G>C variant of CACNG2 (NM_006078.4) was predicted to affect splicing and classified as a likely pathogen. Previously, a provisional association with intellectual disability was proposed based on a single case without epilepsy [MIM #614256] [49]. However, the bi-allelic disruption of cacng2 is well characterized in an animal model such as the stargazer mouse, which presents spike-wave discharges (SWDs), representative of absence seizures in humans [50,51]. Therefore, the association between *CACNG2* alteration and epilepsy is still weak, and its implication in N-type (Cav2.2) is not yet well established.

Summarizing all the above, bi-allelic loss-of-function pathogenic variants in *CACNA1B* have been associated with a complex and progressive neurological syndrome, with epileptic encephalopathy, severe neurodevelopmental delay, and regression, among other features. However, more cases should be collected and reported to fully understand the associated phenotype. As mentioned before, the auxiliary subunits are not specific for N-type calcium channels. In fact, an increasing number of recent studies suggest that, for instance, individual α2δ isoforms exert specific neuronal functions beyond their classical role as calcium channel subunits. Even so, the main phenotypes associated with bi-allelic loss-function variants in *CACNA2D1* and *CACNA2D2* genes are epileptic and developmental encephalopathies, no doubt due to its participation in N- and/or P/Q-type calcium channels. However, the relative importance of one or another channel has, so far, not been disclosed. Finally, the association of other auxiliary subunits, such as *CACNB4* and *CACNG2*, with epilepsy requires more evidence to be definitively established.

### 3.2. N-Type Ca Channel and Epilepsy: Evidences in Animal Models

Using gene knockout mice of particular genes is one of the most effective methods in conducting a successful study on clinical phenotypes, drug discovery, or in preclinical studies. In the case of N-type Ca^2+^ channel studies, and more specifically the *CACNA1B* gene, knockout murine models were applied to examine various clinical consequences (learning and memory, cardiovascular, nociceptive, neurodegenerative); however, its role in epilepsy has never been published [52,53]. On the other hand, disruption of the Cacng2 gene in the stargazer mouse is associated with recurrent epileptic seizures and ataxia. Although this gene codifies a neuronal-specific gamma subunit that may participate in the formation of several types of calcium channels, not only N-type, an in vitro study in Xenopus oocytes demonstrated that this subunit interferes with G protein modulation of N-type calcium channels expressed in Xenopus oocytes [54]. References to other naturally occurring mouse mutant strains for other components of N-type VGCCs, such as ducky, which carried null alleles for *Cacna2d2*, or lethargic, inactivated for the *Cacnb4* gene, have been mentioned above in relation to their epileptic phenotype. However, the subunits coded for these mutated genes are not specific to the N-type VGCCs, so other types of channels might also be implicated.

Other genetic, pharmacological, and electrophysiological data suggest a significant, although poorly understood, role for N-type voltage-dependent calcium channels in epilepsy. Different animal models, mainly in rat and mouse, have allowed analyzing different brain tissues in order to glimpse the implication of VGCCs in epilepsy (Table 5). For instance, partially isolated cortex (“undercut”) is a well-established animal model of posttraumatic epileptogenesis, where dysfunction of Ca^2+^ channels in presynaptic GABAergic terminals via N-type channels plays a prominent role [55]. The Wistar Albino Glaxo Rijswijk (WAG/Rij) rat is a strain that spontaneously develops a clinical phenomenon resembling human absence epilepsy, where excessive calcium entry into the cell, have been proposed to trigger the epileptic activity. The blockage of the N-type calcium channel with conotoxins on this strain decreased the frequency and duration of spike-wave discharges, which let us conclude that N-type calcium channels play an activator role together with T-type VGCCs [56]. By analysis of the evoked epileptiform activity in neocortical slices of rats pointed to the importance of calcium influx mainly via the N-type calcium channel in the epileptiform activity induced in vitro [57]. Amygdala kindling is an experimental model for temporal lobe epilepsy obtained by reiterative electrical stimulations. This stimulation induces a significant increase of N-type VGCCs in the hippocampus, twice in the contra-lateral to the stimulating electrode. The authors argued for the contribution of these channels to seizure-induced synaptic plasticity [58,59]. The increased expression of α1B in both the strata granulosum and the molecular layer of the dentate gyrus after pilocarpine-induced status epilepticus in mice have been suggested to be involved in the occurrence of spontaneously recurrent seizures [60]. In other animal models, the seizure-sensitive gerbil, the elevated expressions of VGCC subtypes (including N-type VGCC) may increase Ca^2+^-dependent excitatory transmission in the hippocampus [61]. Finally, DBA/2J is a mouse inbred strain with high susceptibility to audiogenic seizures. Development of N-type calcium channels in the brain is different in epileptic mice (exposed to auditory stimulation) from nonepileptic mice (not exposed), and this difference was argued to represent a change in synaptic activity that plays a role in epileptogenesis [62].

On the other hand, different studies on physiological conditions related to epilepsy further support the implication of this type of calcium channel. Ethanol withdrawal is one of these conditions, which in rats leads to a significant decrease of the protein levels of α1B subunit, specific for the N-Type VGCC, and this decrease was proposed to secondarily enhance the susceptibility to seizures [63]. Similarly, the loss of the somatostatin function postseizure could result in abnormal synaptic potentiation that contributes to epileptogenesis through N-type Ca^2+^ channels [64]. It is known that crotoxin, a snake venom, causes chronic seizure effects by inducing calcium-dependent glutamate release via N and P/Q calcium channels [65]. Finally, in a study on the potential relationships among aging, calcium channels, and late-onset epilepsy, no overall age-related change in the number of L-type and N-type VGCC in brain areas frequently involved in seizure activity was found. Consequently, age-related changes in brain Ca^2+^ physiology was proposed to be associated with changes in VGCC function rather than the channel number [66].

**Table 5 ijms-24-06100-t005:** Contribution of different animal models to clarify the association of N-type VGCC with epilepsy.

Animal	Tissue/Seizure Type	Effect	Ref.
rat	neocortex	Epileptiform activity in the rat neocortex may occur, at least partially, via the activation of the N-type VGCC.	[57]
mouse	dentate gyrus/CA1 hippocampus	Somatostatin (SST) reduction of dendritic Ca^2+^ through N-type VGCC may contribute to the modulation of synaptic plasticity at long-term potentiation (LTP) synapses. Loss of SST postseizure could contribute to epileptogenesis.	[64]
rat	pyramidal (Pyr) neurons of the sensorimotor cortical	Decreased calcium influx via N-type VGCC in presynaptic GABAergic terminals is a mechanism contributing to decreased inhibitory input onto layer V Pyr cells in this model of cortical posttraumatic epileptogenesis.	[55]
rat	frontal and occipital cortical/absence epilepsy	T and N-type VGCCs play activator roles in spike-wave discharges (SWDs) and have positive effects on the frequency and duration of these discharges.	[56]
rat	inferior colliculus neurons (IC)/ethanol withdrawal seizures	Ethanol withdrawal significantly decreased the protein levels of α1B subunit, specific for the N-Type VGCC, and secondary enhanced the susceptibility to seizures.	[63]
Xenopus	oocytes	Loss of neuronal protein stargazin is associated with recurrent epileptic seizures and ataxia in mice. Stargazin modulates neuronal N-type VGCC by a Gβγ-dependent mechanism.	[54]
rat	cerebral cortex synaptosomes	Crotoxin (Crtx) acts in the CNS, causing chronic seizure effects and other cytotoxic effects. Crtx induces calcium-dependent glutamate release via N and P/Q VGCC.	[65]
rat	hippocampus	Amygdala kindling induced an increase of N-type VGCC expression in the hippocampus.	[58]
rat	hippocampus	N-type VGCC trafficking mechanisms to cause a persistent, local, remodeling of their distributions in CA1 dendrites.	[59]
mouse	neurons and astrocytes/pilocarpine-induced status epilepticus (PISE)	N-type VGCC translocation occurs at acute stages during and after pilocarpine-induced status epilepticus (PISE). The increased expression of this channel in the strata granulosum and dentate gyrus at the chronic stage may be involved in the occurrence of spontaneously recurrent seizures.	[60]
gerbil	hippocampus	The elevated expressions of VGCC subtypes, including N-type VGCC, may increase Ca^2+^-dependent excitatory transmission in the hippocampus of the seizure-sensitive gerbil.	[61]
rat	hippocampus, entorhinal cortex, and neocortex	No overall age-related change in the number of L-type and N-type VGCC in brain areas frequently involved in seizure activity suggests that age-related changes in brain Ca^2+^ physiology may be associated with VGCC function rather than a channel number.	[66]
mouse	brain/audiogenic seizure	Development of N-Type VGCC in the brain is different in epileptic mice (generalized seizures when exposed to auditory stimulation) from nonepileptic mice.	[62]

CNS: central nervous system; GABA: gamma-aminobutyric acid; Ref: references; VGCC: voltage-gated calcium channel.

### 3.3. Anti-Epileptic-Drug Response and Therapeutic Target

The crucial role of calcium ions and glutamate-induced calcium influx via N-methyl-D-aspartate receptors (NMDARs) in neuropathological conditions is described in various studies. The subsequent abnormal NMDAR-dependent calcium accumulation in postsynaptic neurons seems to contribute to a cascade of cellular events that lead to neuronal cell death [67].

Excessive calcium entry into depolarized neurons contributes significantly to neuronal injury. Taking into account that: (i) VGCCs regulate intracellular calcium concentration, essential for several neuronal functions such as cellular excitability, neurotransmitter release, hormone secretion, intracellular metabolism, neurosecretory activity and gene expression, and (ii) N-type Ca channel expression is exclusive for neuronal tissue; it has been proposed that a blockade of N-type VGCCs may be useful for treating neurological diseases, including ischemia, pain, and epilepsy.

Different antiseizure drugs (ASD) may act as N-type channel blockers (Table 6). However, they don’t show the same efficacy against different types of seizures, probably due to their additional mechanisms of action. For many decades carbamazepine was the standard treatment for focal-onset seizures [68], but its use has declined with the marketing of new antiseizure medications that have pharmacokinetic advantages. It may exacerbate absence, myoclonic, and atonic seizures. Gabapentin can be used as an adjunctive treatment for focal seizures. As carbamazepine, it may worsen myoclonus [69]. Both lamotrigine and levetiracetam are broad-spectrum ASD, and they are widely used as first-line ASD for focal seizures and generalized tonic-clonic seizures [70]. Besides, levetiracetam has shown efficacy against myoclonic seizures [71]. Finally, lacosamide is effective against focal-onset seizures as well as generalized onset tonic-clonic seizures [72]. It has not demonstrated efficacy against absences or myoclonic seizures, but it is unlikely to worsen them.

## 4. Conclusions

The evidence so far gathered allows us to be confident about the causality of bi-allelic loss-of-function variants in several components of the N-type voltage-gated calcium channels, coded by *CACNA1B*, *CACNA2D1*, and *CACNA2D2* genes, on epileptic and developmental encephalopathies. On the other hand, further evidence is necessary to define the implication of other components of these channels, such as *CACNB4* or *CACNG2*. Similarly, the role of heterozygous variants in *CACNA2D1* as a predisposing factor to epilepsy and/or intellectual disability remains to be confirmed.

Besides, different animal models, mainly in rat and mouse, have allowed analysis of different brain tissues in order to try to clarify the implication of N-type VGCCs in epilepsy. Even more, distinct ASD may act as N-type channel blockers, and blockade of N-type VGCC has been proposed as a feasible treatment for epilepsy. Therefore, although there is currently strong evidence, future investigations will allow us to finally establish the association of N-type Ca channels with epilepsy syndromes and epilepsy.

## Figures and Tables

**Figure 2 ijms-24-06100-f002:**
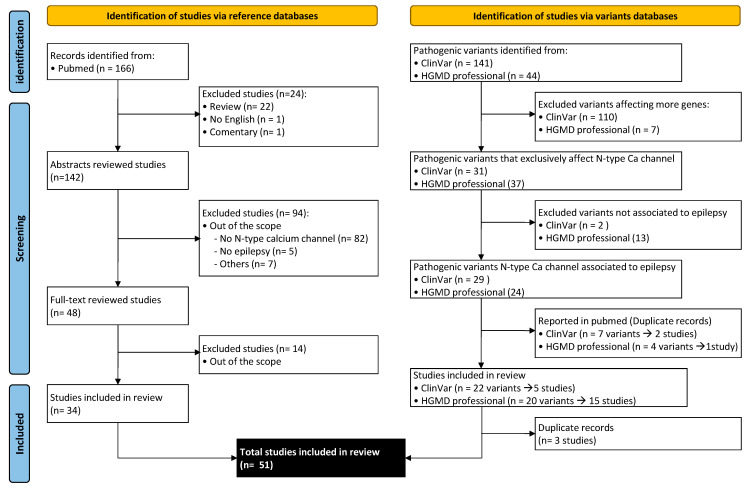
Flow diagram summarizing the identification, screening, and studies selection from PubMed, ClinVar, and HGMD professional base on PRISMA 2020 flow diagram template [23].

**Table 6 ijms-24-06100-t006:** Effect of different drugs on N-type Ca Channels.

Drug	Effect	Model	Tissue/Seizure	Ref.
Lamotrigine (LAG) ^1^	**Inhibition of N-type** and P-type VGCC	Rat	amygdalar and cortical neurons	[73,74,75]
Carbamazepine (CBZ) ^1^	Inhibitory effects on VGCC activity	Rat	frontal cortex	[76]
(S)-4-Carboxy-3-hydroxyphenylglycine ((S)-4C3HPG) ^2^	Induced HVA current inhibition was mediated through the inhibition of group I and activation of group II mGluRs. **Group II mGluRs affected N-type channels**	Human cell	dentate gyms neurons/pharmacoresistant TLE	[77]
Gabapentin (GBP) ^1^	**Induces a significant reduction in the number of N-type functional VGCC in the plasma membrane. Involve GABAB receptor coupling to G-proteins and modulation of** potassium channels and **N-type VGCC**	Cell line Rat	- hippocampal slices	[78,79]
Levetiracetam (LEV) ^1^	**Selective blockers of N-type VGCC**	Rat	CA1 hippocampal neurons cortical neurons	[74,80]
Seletracetam (SEL) ^1^	**Selective blockers of N-type VGCC**	Rat	pyramidal neurons (paroxysmal depolarization shifts)	[81]
(S)-lacosamide (LCM) ^1^	Inhibits CRMP2 phosphorylation culminating in a **reduction of calcium influx via N-type VGCC**	-	-	[82]

^1^ antiseizure drug (ASD), ^2^ agonist of group II mGluRs. **Bold**: effect in N-type VGCC. CRMP2: collapsin response mediator protein 2; GABA: gamma-aminobutyric acid; HVA: high-voltage activated; mGluRs: metabotropic glutamate receptors; Ref: references; TLE: temporal lobe epilepsy; VGCC: voltage-gated calcium channel.

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
