# Peer review of "N-Type Ca Channel in Epileptic Syndromes and Epilepsy: A Systematic Review of Its Genetic Variants"

_ijms, 2023, doi:10.3390/ijms24076100_

Round 1
Reviewer 1 Report
The submitted review accumulates the data about the role of N-type voltage-gated calcium channels in the epilepsy. Great attention has been paid to the data demonstrating correlation between mutations in the genes encoding N-type VGCC subunits and epilepsy development. I have only some minor recommendations.
- Please, explain "CNVs" abbreviation;
- The biological effects of some drugs in Table 5 should be revised. For instance, it is known that levetiracetam and seletracetam also realize their antiepileptic effects via interaction with synaptic vesicle protein or glycine receptor subunits whereas the authors report only the effect on N-type calcium channels ignoring other activities.
- Paragraphs in section 3.1 end with short conclusions, while no conclusions have been drawn in section 3.2.
Author Response
We appreciated your constructive criticisms. We have addressed each of your concerns as outlined below.
1) Please, explain "CNVs" abbreviation.
The meaning definition of CNV has been introduced in the text.
2) The biological effects of some drugs in Table 5 should be revised. For instance, it is known that levetiracetam and seletracetam also realize their antiepileptic effects via interaction with synaptic vesicle protein or glycine receptor subunits whereas the authors report only the effect on N-type calcium channels ignoring other activities.
We have focused exclusively on the effect on N-type calcium channels, as specified in the aims of the review.
3) Paragraphs in section 3.1 end with short conclusions, while no conclusions have been drawn in section 3.2.
Section 3.1 has been modified to include a final conclusion and section 3.2 has been totally rewritten. We hope that these changes substantially improve the quality of the paper and fulfill the reviewer requirements.
Reviewer 2 Report
This is an interesting review article that deserves publication after a minor change. Only few points should be addressed:
1) The Authors should read the whole text thoroughly to find and correct small errors in the English style. I would suggest to get help form an English native speaker.
2) What is the meaning of the symbol alpha1B subunit (Figure 1). Is this the same as the alpa1 subunit ?
3) The Authors should explain the meaning of the terms: high-voltage activated and low-voltage activated. What is "high voltage" and what is "low voltage" ?
4) The Authors should provide more informaion about biophysical and pharmacological properties of the "N" ("normal") voltage-gated potassium channels in relation to best-known types of these channels, such as the "L" ("large") and the "T" ("transient") voltage-gated calcium channels.
5) Is it known how the mutations influence the activity of the channels, such as the magnitude of calcium current carried, the channel activation and inactivation ?
Author Response
We appreciated your constructive criticisms. We have addressed each of your concerns as outlined below.
1) The Authors should read the whole text thoroughly to find and correct small errors in the English style. I would suggest to get help form an English native speaker.
English has been reviewed and corrected.
2) What is the meaning of the symbol alpha1B subunit (Figure 1). Is this the same as the alpa1 subunit ?
Yes, α1B refers to α1 subunit of the Cav2.2 channel. However, since it is a schematic representation of the structure of the VGCC, the figure 1 has been corrected to “α1”.
3) The Authors should explain the meaning of the terms: high-voltage activated and low-voltage activated. What is "high voltage" and what is "low voltage" ?
Done
4) The Authors should provide more information about biophysical and pharmacological properties of the "N" ("normal") voltage-gated potassium channels in relation to best-known types of these channels, such as the "L" ("large") and the "T" ("transient") voltage-gated calcium channels.
A new table (Table 2) has been added to include this information in the review.
5) Is it known how the mutations influence the activity of the channels, such as the magnitude of calcium current carried, the channel activation and inactivation ?
All the undoubtedly pathogenic variants reported in the paper are loss-of-function mutations that presumably lead to inactivation of the channels, mainly frameshifts and splicing. Missense variant p.L1040P in the CACNA2D2 gene also was proven to result in reduced current density and slow inactivation in both N-type (Cav2.2) and L-type (Cav1.2) calcium channels (page 8, lines 215-218). The pathogenicity of other missense variants, such that in CACNB4, is unclear.
We expect that these modifications fulfill the reviewer requirements.
Reviewer 3 Report
Comments are listed here for the author’s consideration to further improve the quality and overall impact of the manuscript.
1. Title should include EPILEPTIC SYNDROMES and epilepsy.
2. Line 19. “Therefore, the present review systematically summarizes existing publications regarding the genetic associations between N-type voltage-dependent calcium channel and epilepsy.” And where is the preclinical information? Isn't that the goal of this review? Authors must include this information.
3. The introduction does not mention anything regarding epilepsy or epilepsy and calcium channels. Authors must include this information.
4. Line 31: Include a reference.
5. The information of genetic variants of auxiliary subunits, mentioned in this review (CACNA2D1, CACNA2D2, CACNB4, CACNG2) are specific to N-type calcium channels? Or is general information respect to calcium channels? It is not clear in the manuscript.
6. Methods. Clarify that studies with both clinical and preclinical information were included for this review.
7. Results and discussion. In general authors just describe the studies consulted and do not give any discusión in each subsection. Correct it.
8. Results and discussion. Change “3.1 Case reports” by 3.1. N-type Ca channel and epilepsy: clinical evidence.
9. Section 3.1.1. Discussion is missing. Authors should include it.
10. Line 195-199. “Homozygous knockout of Cacna2d1 in mice…” this information is not relevant. Remove this paragraph.
11. Line 298. “…the implication of VGCCs in epilepsy.” Change it by “….N-type VGCCs in epilepsy.”
12. Sections 3.2 and 3.3. must be described by the authors, since they only mention tha tables 4 and 5, and do not explain the studies consulted and the main findings. Also a discussion is missing. Correct it.
13. Remove table 6, since it is not important information concerning N-type VGCCs which is the main goal of this review.
14. Line 322 “Different antiepileptic drugs (AED)…” change it by antiseizure drugs (ASDs).
15. Authors should discuss how these findings about of these genetic variants could help to generate new tratments or improve the existing treatments.
16. Authors should discuss and mention the relevance of preclinical findings and available ASDs that block N-type VGCCs, for the treatment of seizures and epilepsy. If this drugs have antiseizure efficacy or not, and how the modulation of different subunits of N-type VGCCs could help for the treatment.
Author Response
We appreciated your constructive criticisms. We have addressed each of your concerns as outlined below.
1) Title should include EPILEPTIC SYNDROMES and epilepsy.
The title has been modified to “N-Type Ca channel in epileptic syndromes and epilepsy”. We appreciate the suggestion.
2) Line 19. “Therefore, the present review systematically summarizes existing publications regarding the genetic associations between N-type voltage-dependent calcium channel and epilepsy.” And where is the preclinical information? Isn't that the goal of this review? Authors must include this information.
The goal of the present review was to gather the information on the relationship between the N-type voltage-dependent calcium channel, and more specifically of the genetic variants of the different components of these channels, with epilepsy. In any case, Table 5 (now table 6) gathers all the available information on animal models and in vitro preclinical information.
3) The introduction does not mention anything regarding epilepsy or epilepsy and calcium channels. Authors must include this information.
The association of some voltage-gated calcium channel components with epilepsy has been widely reported and it was mentioned in lines 41 to 49 of the introduction.
4) Line 31: Include a reference.
Done
5) The information of genetic variants of auxiliary subunits, mentioned in this review (CACNA2D1, CACNA2D2, CACNB4, CACNG2) are specific to N-type calcium channels? Or is general information respect to calcium channels? It is not clear in the manuscript.
As specified in the last paragraphs of Introduction (see page 4), the auxiliary subunits mentioned in this review are not specific to N-type calcium channels. In fact, an increasing number of recent studies suggests that, for instance, individual α2δ isoforms exert specific neuronal functions beyond their classical role as calcium channel subunits. However, the main phenotype associated to biallelic loss-function variants in CACNA2D1, CACNA2D2 genes are epileptic and developmental encephalopathies, no doubt due to its participation in N- and/or P/Q-type calcium channels. However, the relative importance of one or another channel is so far not disclosed.
6) Methods. Clarify that studies with both clinical and preclinical information were included for this review.
It has been clarified in the section of Methods as suggested.
7) Results and discussion. In general authors just describe the studies consulted and do not give any discussion in each subsection. Correct it.
We are sorry to disagree. Several case reports on the putative causal variants in several auxiliary subunits have been doubted in the respective sections on the gene in question, based on other currently available results (model animals, aggregated genomic databases, etc.). However, it has been modified to include a better discussion of each subsection.
8) Results and discussion. Change “3.1 Case reports” by 3.1. N-type Ca channel and epilepsy: clinical evidence.
It has been modified as suggested.
9) Section 3.1.1. Discussion is missing. Authors should include it.
Section 3.1 has been modified to include a final discussion, as well as Section 3.1.1.
10) Line 195-199. “Homozygous knockout of Cacna2d1 in mice…” this information is not relevant. Remove this paragraph.
Thanks for this comment. We agree it is out of the scope of the review; in fact, a very recent addition to OMIM (* 114204; modified on February 8, 2023) rules out the alleged relationship with short QT syndrome.
11) Line 298. “…the implication of VGCCs in epilepsy.” Change it by “….N-type VGCCs in epilepsy.”
Done
12) Sections 3.2 and 3.3. must be described by the authors, since they only mention the tables 4 and 5, and do not explain the studies consulted and the main findings. Also a discussion is missing. Correct it.
Sections 3.2 and 3.3 have been completed. We hope that these changes substantially improve the quality of the paper and fulfill the reviewer expectation.
13) Remove table 6, since it is not important information concerning N-type VGCCs which is the main goal of this review.
Table 6 has been removed as suggested.
14) Line 322 “Different antiepileptic drugs (AED)…” change it by antiseizure drugs (ASDs).
Done.
15) Authors should discuss how these findings about of these genetic variants could help to generate new treatments or improve the existing treatments.
This is the great pending expectation, but on which unfortunately we have not found any published information. It would not be serious to speculate on our part about it.
16) Authors should discuss and mention the relevance of preclinical findings and available ASDs that block N-type VGCCs, for the treatment of seizures and epilepsy. If this drugs have antiseizure efficacy or not, and how the modulation of different subunits of N-type VGCCs could help for the treatment.
Lamotrigine, carbamazepine, gabapentin, levetiracetam and lacosamide are well-known ASD. Their efficacy against different types of seizures has been included in section 3.3.
We expect that these modifications fulfill the reviewer requirements.
Round 2
Reviewer 3 Report
Title should include genetic variants “N-type Ca channel in epileptic syndromes and epilepsy: a Systematic Review of its genetic variants”.
Author Response
The title has been modified as suggested